∂ | **Open Peer Review** | Virology | Research Article

# Structural and functional analyses of SARS-CoV-2 Nsp3 and its specific interactions with the 5' UTR of the viral genome

Sofia Lemak,[1] Tatiana Skarina,[1] Deepak T. Patel,[2] Peter J. Stogios,[1] Alexei Savchenko[1,2,3]

**ABSTRACT** Non-structural protein 3 (Nsp3) is the largest open reading frame encoded in the SARS-CoV-2 genome, essential for the formation of double-membrane vesicles (DMV) wherein viral RNA replication occurs. We conducted an extensive structure-function analysis of Nsp3 and determined the crystal structures of the ubiquitin-like 1 (Ubl1), nucleic acid binding (NAB), β-coronavirus-specific marker (βSM) domains, and a sub-region of the Y domain of this protein. We show that the Ubl1, ADP-ribose phosphatase (ADRP), human SARS Unique (HSUD), NAB, and Y domains of Nsp3 bind the 5' UTR of the viral genome and that the Ubl1 and Y domains possess affinity for recognition of this region, suggesting high specificity. The Ubl1-Nucleocapsid (N) protein complex binds the 5' UTR with greater affinity than the individual proteins alone. Our results suggest that multiple domains of Nsp3, particularly Ubl1 and Y, shepherd the 5' UTR of the viral genome during translocation through the DMV membrane, priming the Ubl1 domain to load the genome onto N protein.

**IMPORTANCE** The largest protein encoded by the SARS-CoV-2 genome is Nsp3. In infected cells, this multi-domain protein forms a pore structure in the virus-induced double-membrane vesicles (DMV). We have incomplete data on Nsp3 molecular structure, and here, we describe crystal structures for multiple domains of Nsp3. It is thought that newly replicated viral RNA transits through the DMV pore; however, we possess incomplete data on which regions of Nsp3 actually interact with RNA. Here, we present data showing that five domains of Nsp3 interact with the 5' UTR of the SARS-CoV-2 RNA, including the Y domain for which no function has ever been discovered. These data suggest that the pore structure plays an active role in recognizing the terminal end of the genome, transiting and loading the viral RNA onto the cytoplasmic nucleocapsid protein. These data help expand our knowledge of Nsp3 structure and function and the SARS-CoV-2 replication cycle.

**KEYWORDS** SARS-CoV-2, virology, structural biology, RNA binding , protein-protein interactions

The COVID-19 pandemic has brought into focus the danger and complexity of viral infections. As of the time of writing, SARS-CoV-2 caused more than 7 million deaths from more than 704 million infections (Worldometers.info). Research and development into direct-acting antivirals highlighted the necessity of a detailed molecular understanding of the mechanisms of viral pathogenesis and the host-pathogen interactions that could be intercepted by antiviral therapeutics. In this vein, structural biology approaches have delivered stunning and rapid successes in providing a molecular understanding of the SARS-CoV-2 proteins and their interactions with host factors (reviewed in [1, 2]). However, much remains to be learned about the molecular structure and function of some of this virus' open-reading frame products and their interactions with host proteins.

Address correspondence to Peter J. Stogios, p.stogios@utoronto.ca, or Alexei Savchenko, alexei.savchenko@ucalgary.ca.

The authors declare no conflict of interest.

See the funding table on p. 14.

The SARS-CoV-2 virus's genome represents a 30 kb single-stranded positive-sense RNA encapsulated by the nucleocapsid (N) protein. The virion is protected by a host-derived membrane envelope harboring the Spike (S), Membrane (M), and Envelope (E) structural proteins. The SARS-CoV-2 genome encodes a total of 14 open reading frames, which translate into 29 viral proteins. *orf1a* encodes the polyprotein pp1a, which is processed by the papain-like protease (PlPro) itself encoded within this open reading frame, into non-structural proteins (Nsps) 1–3. The genome also encodes *orf1b,* and a −1 ribosomal frameshift upstream of the sequence corresponding to *orf1a's* stop codon results in readthrough into Orf1b, which translates the polyprotein Orf1ab. Orf1ab encodes Nsps 4–10, which are liberated by the Main protease/Nsp5 region or Orf1ab.

Spanning 1,945 residues, Nsp3 is the largest single protein encoded by the virus (reviewed in [3]). Along with Nsp4 and Nsp6, Nsp3 mediates the formation of double-membrane vesicles (DMVs) in infected cells, with the C-terminal portion (600 residues) of Nsp3 shown to be essential for this activity (4, 5). These virus-induced organelles are rich in double-stranded RNA. Accordingly, the DMVs are suggested to contain the viral replication-transcription (RTC) complex, shielding it from cytoplasmic RNA sensors that activate the innate immune system (6–10). Cryo-electron tomography (cryo-ET) studies of cells infected with SARS-CoV-2 or murine hepatitis virus (MHV) show that Nsp3 localizes to molecular pore structures embedded in the DMV membrane (9–12). Since the N protein is localized in the cytoplasm of infected cells (10), nascent viral ssRNA genomes must exit the DMV for packaging into a ribonucleoprotein complex with N. Similarly, viral mRNAs transcribed from the genome within DMVs must exist in this organelle for translation by cytoplasmic/ER-associated ribosomes. Therefore, the Nsp3-containing molecular pore is thought to provide a key gate between viral RNAs in the DMV lumen and the cytoplasm, which facilitates Nsp3-RNA and Nsp3-N protein interactions to facilitate RNA exit and packaging (11–20).

Nsp3 from SARS-CoV-2 is comprised of at least seven structural domains (Fig. 1a). Although PlPro encoded as part of this protein has been the focus of intensive research as an established target of antiviral therapies, little is known about the molecular structure and function of other Nsp3 domains. A comprehensive study of the RNA-binding properties of SARS-CoV-2 Nsp3 throughout the domains of this protein has not been carried out.

After PlPro, the next best-characterized region of Nsp3 is the ADP-ribose phosphatase (ADRP) domain (also known as the macrodomain or domain X). ADRP has been experimentally shown to harbor ADP-ribose phosphatase activity, has been the subject of numerous inhibitor screening campaigns (21–24), and has been shown to remove ADP-ribosyl groups from host PARP14 (25). Accordingly, this domain has been structurally characterized in complex with various ligands including small molecule inhibitors (24, 26, 27).

In contrast to ADRP, which appears to lack RNA binding activity, the Ubl1, human SARS unique (HSUD), and NAB domains of SARS-CoV Nsp3 have been shown to possess this activity (19, 20, 28–30). The Ubl1 domain of SARS-CoV Nsp3 was reported to have 20 µM affinity for ssRNA (20). It has also been predicted that the binding sequence for SARS-CoV Nsp3 Ubl1 is 5′-AUA-′3, based on its co-purification with AUA-containing RNA, and biochemical characterization of short oligonucleotide binding (20). The HSUD domain of SARS-CoV and SARS-CoV-2 Nsp3 has been shown to bind G-quadruplex sequences (18, 28, 30, 31). The SARS-CoV Nsp3 NAB domain possesses nucleic acid-binding activity at micromolar concentrations to both ssDNA and RNA, with a preference for RNA substrates (32), and has been shown to bind to short, G-rich ssRNA, specifically those with three consecutive G residues (20).

A critical role of the Nsp3 Ubl1 domain in viral RNA synthesis has been attributed to its interaction with the N protein, which tethers Nsp3 to viral RNA during replication (16, 17, 33, 34). Genetic interaction assays have shown that the N-terminal region of MHV Nsp3, which contains Ubl1, binds to MHV N protein in an RNA-independent and species-specific manner (14, 35). In SARS-CoV-2, it has been shown that the Nsp3 Ubl1 domain

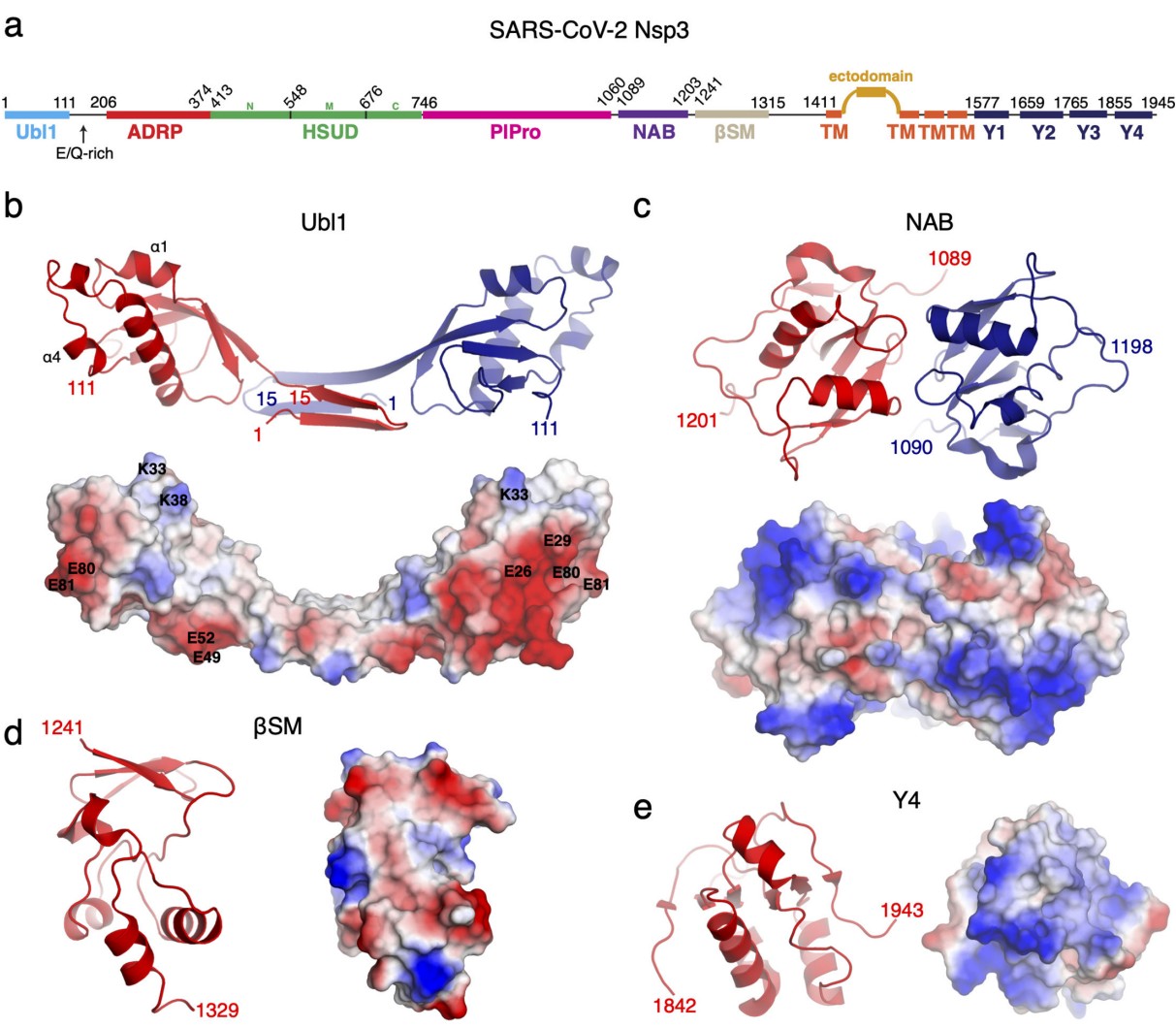

**FIG 1** (a) Domain architecture of the SARS-CoV-2 Nsp3 protein. Numbers above the schematic indicate domain boundaries. (b) Crystal structure of Nsp3 Ubl1 domain. (c) Crystal structure of Nsp3 NAB domain. (d) Crystal structure of Nsp3 βSM domain. (e) Crystal structure of Nsp3 Y4 region. For (b–e), shown in cartoon and electrostatic surface representation, where red = negatively charged, white = neutral, blue = positively charged, and numbers indicate N- and C-termini resolved in each crystal structure.

interacts with the N-terminal domain of the N-protein as well as with two regions in the linker region between its N-terminal and C-terminal domains (34, 36). Since both proteins independently bind RNA and given the low occurrence of sequences predicted to be recognized by Ubl1 in the 5′-UTR or 3′-UTR, it has been speculated that the RNA-binding properties of Ubl1 have a role in its connection to N protein (33).

The final two domains of Nsp3 are the beta-coronavirus-specific marker (βSM) domain (also known as β2M) and the so-called Y domain, which localizes to the extreme C-terminus of Nsp3 (33). The Y domain has been subdivided into four regions: Y1, Y2, Y3, and Y4 (37), with Y2 and Y3 regions restricted to *Coronaviridae* (32) and the Y2 and Y4 regions sharing higher conservation. The function of the Y domain of Nsp3 remains unknown.

In this study, we focused on expanding our understanding of the molecular features and function of the domains of SARS-CoV-2 Nsp3 outside of the PlPro domain, with a focus on interactions with viral RNA and the N protein. We determined the crystal structures of the Ubl1, NAB, βSM, and Y4 domains of Nsp3, the last two of which provided the first experimentally derived molecular images of these regions of Nsp3. We show that five domains of SARS-CoV-2 Nsp3 interact with the 5′ UTR of the viral

genome, including Ubl1, HSUD, ADRP, NAB, and the Y domain. We demonstrate that the Ubl1-N protein complex shows higher affinity for the 5′ UTR than the isolated proteins, suggesting synergy between Ubl1's N protein binding and RNA recognition. We also show that the Y domain possesses an affinity to RNA, a function never previously attributed to this region of Nsp3. Altogether, these findings indicate that multiple regions of Nsp3 play important roles in shepherding the 5′ end of the viral genome through the DMV membrane for loading onto the N protein and suggest that they line the interior surface of the DMV pore.

## RESULTS

### Crystal structures of the Nsp3 Ubl1, NAB, βSM, and Y4 domains

As a first step in our functional analysis of Nsp3, we pursued structural characterization of the individual domains in this protein. Ubl1 (residues 1–111), ADRP (residues 206–374), HSUD (residues 413–676), NAB (residues 1,050–1,216 and residues 1,089–1,203), and the βSM (residues 1,230–1,334) were recombinantly expressed and purified from *E. coli*. Along the same lines, we expressed and purified the full Y domain (residues 1,584–1,945) and two fragments containing the Y2–Y4 region and the Y4 region alone (1,619–1,945 and 1,844–1,945) (Fig. 1a; Fig. S1). Using these purified Nsp3 fragments, we were able to obtain structure determination quality crystals for Ubl1, NAB, and the Y4 region (residues 1,844–1,945) of the Y domain (see Materials and Methods for details, and Table S1 for x-ray crystallographic statistics). Notably, after depositions of the Ubl1 domain and the Y4 region structures to the publicly available database (PDB 7KAG, 7TI9, and 7RQG), the structures of the Ubl1-N protein complex and Y2–Y4 region of the Y domain were reported (34, 37).

 The crystal structure of Ubl1 was solved by Molecular Replacement using the previously determined structure of the corresponding domain from the SARS-CoV virus (20) (see Table S1 for crystallographic statistics). We determined the structure of this domain in two crystal forms (form 1 and form 2), both of which showed the same conformation of the Ubl1 protomer, suggestive of the oligomerization architecture of the domain. In the case of the form 1 Ubl1 structure (Fig. 1b), we were able to unambiguously assign all 111 residues of this fragment with two protomer chains present in the asymmetric unit. In contrast, the form 2 Ubl1 structure contained only one polypeptide chain in the asymmetric unit. However, the crystal symmetry of form 2 produced a dimeric structure identical to the dimeric structure in the asymmetric unit of form 1. The RMSDs between the one chain in Ubl1 form 2 and the two chains of Ubl form 1 are 0.23 Å and 0.33 Å over 100 or 101 Cα atoms. Observing the same dimeric arrangement of Ubl1 fragments in two different crystal forms implied this to be functionally relevant for this domain of Nsp3. We observed the same oligomerization state for Ubl1 in size exclusion chromatography (Fig. S1b). The dimerization interface in the crystal structures of Ubl1 was mediated by swapping of two N-terminal β-strands formed by residues 3–15. The crystal symmetry observed in the form 2 structure showed further association of two dimeric Ubl1's into a tetramer, via an extension of this N-terminal β-sheet (Fig. S2a). To test the role of this region in oligomerization, we designed a purified Ubl1 fragment, missing 14 N-terminal residues (Ubl1$^{\Delta 1-14}$). Based on size exclusion chromatography, this fragment remained predominantly monomeric in solution (Fig. S1c) in line with the observed role of the N-terminal region in Ubl1 oligomerization. In line with significant sequence conservation between the SARS-CoV-2 Nsp3 Ubl1 domain with the Nsp3 Ubl1 domains from SARS-CoV and MHV (sharing 75% and 31% sequence identity, correspondingly), these structures superimposed with RMSD of 2.7 Å and 4.1 Å, over 102 and 100 Cα atoms, respectively (Fig. S2b). This overall similarity is broken at the N-termini of the Ubl1 domains: this region in Nsp3s from SARS-CoV and MHV shown to be monomeric in solution and does not adopt the two β-strand arrangement observed in Ubl1 from SARS-CoV-2 (20, 35). Our analysis of the electrostatic surface also revealed a clear acidic patch on one face of the Ubl1 from SARS-CoV-2, a neutral patch on the "top" face of the

Ubl1, and that the central domain-swapped region harbored largely neutral amino acids (Fig. 1b).

The crystal structure of the NAB domain (Fig. 1C) was solved by MR using the structure of the corresponding domain (residues 1,089–1,201) from SARS-CoV Nsp3 (19) (Table S1). We observed two polypeptide chains, each corresponding to the NAB sequence in the asymmetric unit. However, size exclusion chromatography showed that this domain is predominantly monomeric in solution (Fig. S1d). In contrast, the longer fragment of the NAB, comprising residues 1,050–1,216, was dimeric under similar conditions (Fig. S1e), suggesting an additional 15 residues at the N-termini of this domain mediate dimerization. Analysis by PDBePISA on the 1,089–1,201 crystal structure identified the buried surface area between protomers is 1,370 $\text{Å}^3$ but does not predict a stable dimer. Further analysis of the NAB is required to advance our understanding of the functional relevance of dimerization in the NAB, as ours remains the only experimentally derived structure of this portion of Nsp3 currently available. The determined structure of the NAB domain is highly similar to that of the corresponding domain of Nsp3 from SARS-CoV (RMSD 0.9 Å over 113 matching Cα atoms). Importantly, the structures of NAB domains from these two viruses share common features in the positioning of the residues K75/K74, K76/K75, K99/K98, and R106/R105 shown to contribute to RNA binding (19). However, we also observed that the conformations of N- and C-termini differ between the structures of NAB from SARS-CoV and SARS-CoV-2 (Fig. S2c). This difference may be due to these regions serving as flexible linkers to the PlPro and βSM domains, regions mediating dimerization, or differing crystal lattice packing.

The βSM domain of Nsp3 does not share significant primary sequence similarity with any structurally characterized proteins. Therefore, we used AlphaFold2 (38) to generate a model of this domain and used it to solve βSM crystal structure by MR (Table S1). The obtained structure is comprised of a three-stranded β-sheet and short helices packing against the sheet spanning residues 1,241–1,329 of Nsp3 (Fig. 1d). Notably, a search for structurally similar proteins to the βSM structure did not reveal any hits in the PDB database. The asymmetric unit contained 16 copies of the βSM domain. However, the size exclusion chromatography (Fig. S1f) suggested this fragment to be predominantly monomeric in solution in line with PDBePISA server prediction of observed contacts between individual protomers in the crystal lattice, which was not consistent with stable oligomerization. Our analysis of the βSM structure did not reveal any significant clefts or pockets that may be indicative of its molecular function.

Although we were unable to obtain crystals of the full-length Y domain, we were successful in determining the crystal structure of the fragment corresponding to its Y4 region. As in the case of the βSM domain, the structure of the Y4 fragment was determined by MR using an AlphaFold2-generated model (Table S1). Retrospective analysis showed the AlphaFold2 model of the Y4 region closely matched its crystal structure with RMSD 0.4 Å over 84 matching Cα atoms (Fig. S3a). Furthermore, the structure of the corresponding fragment in a consequently determined structure of the Y2 region of the Y domain (PDB 8F2E [37]) also matched our Y4 region structure with an RMSD of 0.5 Å over all 93 matching Cα atoms in this fragment. The Y4 crystal structure featured a mixed α/β structure centered on a central 6-stranded anti-parallel β-sheet (Fig. 1e). Four chains were observed in the asymmetric unit of the Y4 fragment crystal lattice, with intermolecular disulfide bonds formed via Cys1926 in each of two protomer pairs. However, both the Y4 fragment and full-length Y domain remained monomeric in solution according to size exclusion chromatography (Fig. S1g and h), suggesting that the observed arrangements and covalent bonding between the protomers were a consequence of crystal packing and oxidation during the crystallization process, respectively. The Y4 fragment's structure displays positively charged patches on its surface (Fig. 1e). A structural similarity search vs. the PDB showed that the Y4 domain shows only very distantly similar matches (Fig. S2d); this lack of strongly structurally similar proteins suggested by our analysis is in agreement with that done using the structure of full-length Y domain (37).

## Multiple domains of Nsp3 bind the 5′-UTR of the SARS-CoV-2 genome in a specific manner

Previous studies have demonstrated that the domains of Nsp3 from MHV and SARS-CoV bind ssRNA (20, 35). Thus, we explored the presence of such activity for domains of Nsp3 from the SARS-CoV-2 in a comprehensive manner. Previous work (20) suggested that SARS-CoV Nsp3 Ubl1 possesses the affinity for AUA sequences, which appear multiple times in this virus' 5′ UTR (bases 1–231). By analogy, we selected a similar region of the 5′ UTR of SARS-CoV-2 (bases 1–245) for analysis of RNA binding; this region contains three AUA sequences. This region comprises stem loops 1–4 (SL1–SL4) and a portion of SL5 of the 5′ UTR (39). Electrophoretic mobility shift assay (EMSA) using [$^{32}$P]-labeled ssRNA comprising this region of the SARS-CoV-2 5′ UTR showed that the Ubl1 domain of Nsp3 binds this RNA fragment with $K_D$ value of 31 ± 3.9 µM, which is comparable with the affinity established for the corresponding domain from SARS-CoV Nsp3 (20) (Fig. 2a). To determine which charged residues were most significant for ssRNA binding, we used our crystal structure of Ubl1 to guide site-directed mutagenesis to alter negatively charged patches on the surface of this domain to positively charged and vice versa (Fig. 1b, Fig. 2b). The resulting Ubl1 variants were tested for RNA binding to the 5′-UTR substrate in comparison with the wild-type Ubl1 (Fig. 2B). The Ubl1$^{K33EK38E}$ variant carrying substitutions of residues located on the α1-helix (Fig. 2B) showed complete loss of ssRNA binding (Fig. 2B). The Ubl1$^{D80RE81R}$ variant with substituted negatively charged residues at the α4-helix to positively charged ones showed reduced affinity for ssRNA compared with the wild-type Ubl1. In contrast, the Ubl1$^{E26RE29R}$ and Ubl1$^{E49RE52R}$ variants showed ssRNA binding comparable with that of the wild-type Ubl1. Interestingly, the Ubl1$^{Δ1-14}$ deletion variant, which we showed was unable to dimerize, also showed complete loss of binding to 5′ UTR (Fig. S3a), whereas Ubl1$^{E26RE29R}$, Ubl1$^{K33EK38E}$, Ubl1$^{E49RE52R}$, and Ubl1$^{D80RE81R}$ variants remained predominantly dimeric similarly to the wild-type as verified by size exclusion chromatography (Fig. S6 for two representative variants). This observation prompted us to suggest that dimerization is important for Ubl's RNA binding.

EMSA analysis against the 5′-UTR ssRNA 200 residue fragment showed that the ADRP, HSUD, NAB, and Y (residues 1,584–1,945) domains of Nsp3 also show affinity to this substrate (Fig. 3). The calculated $K_D$ values for these domains were 198 ± 17, 41 ± 4, 204 ± 30, and 0.7 ± 0.1 µM, respectively. These affinity values were lower than for Ubl1 with the notable exception of the Y domain, which demonstrated significantly stronger binding. Interestingly, the EMSA assay with the same substrate for the Y4 fragment did not reveal any binding (Fig. S3b), suggesting the important role played in this activity by the portion of the Y domain corresponding to the Y1, Y2 and Y3 regions that were missing in the tested fragment.

To characterize whether Nsp3 binds RNA in a sequence-specific manner, we tested whether the individual domains bind a region immediately downstream from the previously tested region of the 5′ UTR corresponding to 301–545 bases of the SARS-CoV-2 genome. We did not observe binding to this RNA substrate in cases of Ubl1, Ubl1$^{Δ1-14}$, NAB, or Y domains of Nsp3. However, the HSUD domain did show binding to this RNA fragment (Fig. S3c). These results indicate that multiple Nsp3 domains specifically recognize the first 245 bases of the 5′ UTR, with HSUD possessing more promiscuous RNA binding activity.

## The Ubl1+N protein complex binds the 5′ UTR with higher affinity than the proteins alone

Since Ubl1 and N have been shown to form a complex involving the N-terminal domain and linker regions of the N protein (34, 36), we investigated how their interaction affected their binding to this RNA substrate using EMSA. According to our results, the N protein binds to the 5′ UTR with calculated $K_D$ of 0.79 ± 0.11 µM (Fig. 2C). In comparison to the $K_D$ calculated for the binding of Ubl1 domain alone to the same substrate (see above), this value suggests tighter binding (Fig. 2A). The EMSA assay

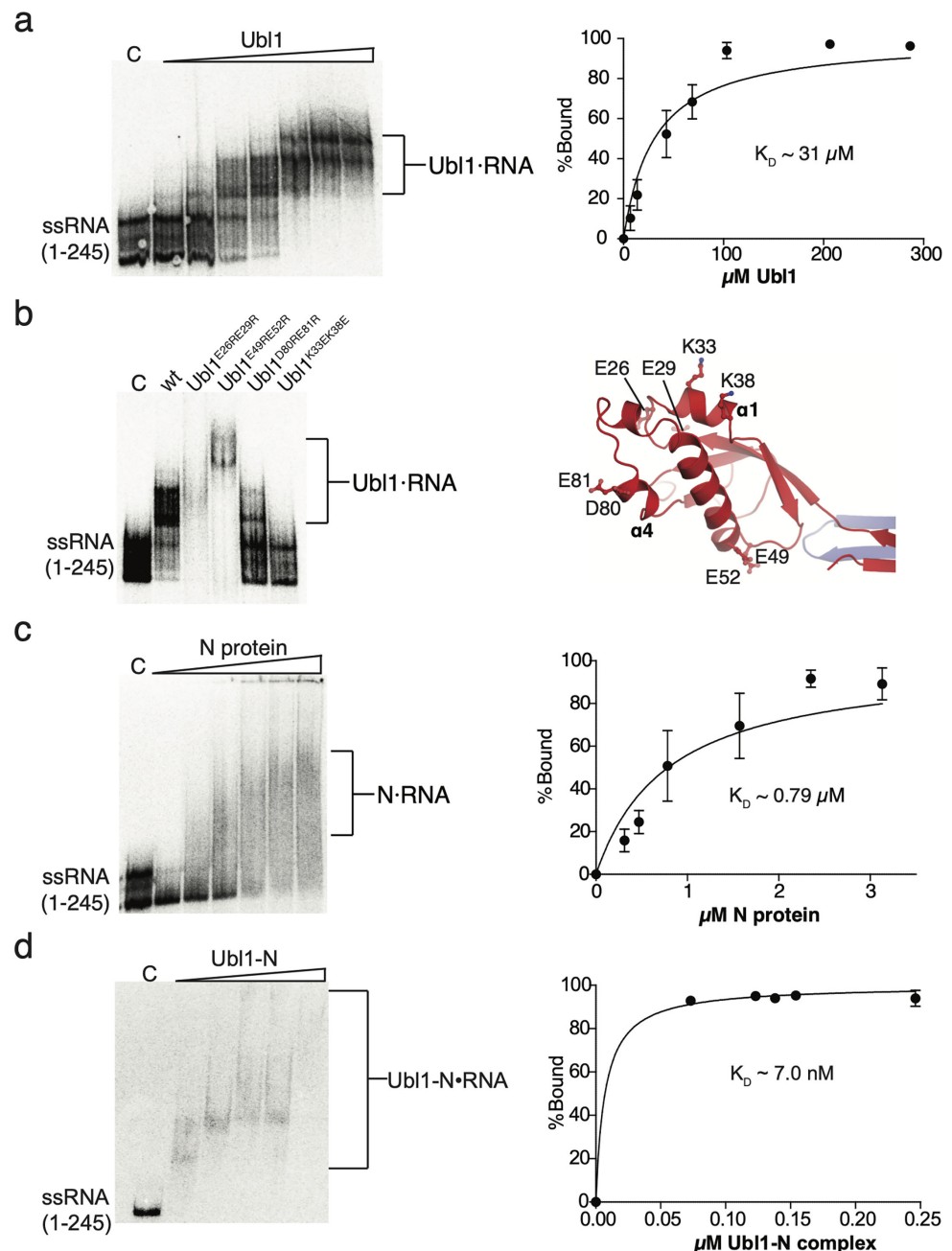

**FIG 2** (a) EMSA and a quantified binding curve showing Nsp3 Ubl1 binding to 5′ UTR (1-245). Subsequent lanes indicate an increasing concentration of Nsp3 Ubl1 (7-287 μM). (b) EMSA showing Nsp3 Ubl1 variants (58 μM each) binding to 5′ UTR (1-245). Right = location of substituted amino acids on the crystal structure of Nsp3 Ubl1 domain. (c) EMSA and quantified binding curve showing 313 nM–3.1 μM N protein binding to 5′ UTR (1–245). (d) EMSA and quantified binding curve showing 69-230 nM Ubl1-N complex binding to 5′ UTR (1-245). For all figures, the lane labeled "C" refers to the control lane with radiolabeled ssRNA only. Binding curves (shown as a mean ± standard deviation from three different gels using the same protein batches) and estimated $K_D$ values were determined by EMSAs using three independent experiments.

against the same substrate using the His$_6$-Ubl1 and N protein complex shows even higher affinity, reflected in calculated $K_D$ of 7.0 ± 0.8 nM using an estimated 4:4 complex as suggested by the SEC-RALS analysis (Fig. 2D; Fig. S4). This result shows that although both the N protein and the Ubl1 domain of Nsp3 demonstrate significant affinity toward the 5′ UTR, the binding is dramatically strengthened by the formation of a complex

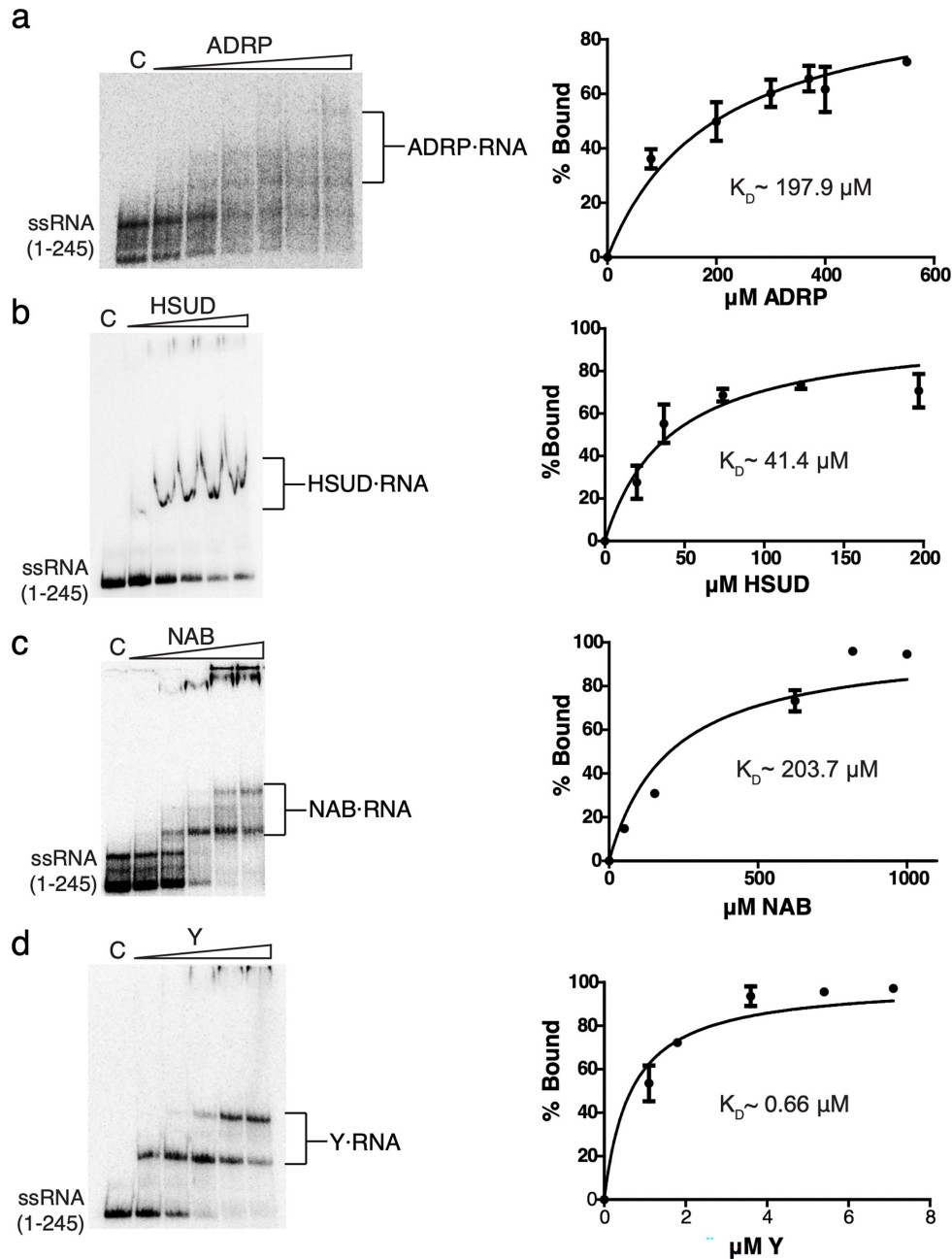

**FIG 3** EMSA and binding curves showing Nsp3 (a) 80–550 µM ADRP, (b) 20–197 µM HSUD, (c) 51 µM–1 mM NAB, and (d) 1.1–7.1 µM Y domains binding to 5′ UTR (1-245). For all figures, the lane labeled "C" refers to the control lane with radiolabeled ssRNA only. For all figures, the lane labeled "C" refers to the control lane with radiolabeled ssRNA only. Binding curves (shown as a mean ± standard deviation from three different gels using the same protein batches) and estimated $K_D$ values were determined by EMSAs using three independent experiments.

between these proteins. The Ubl1/N protein complex was also able to interact with 301–545 bp fragment downstream from the 5′ UTR (Fig. S3c). Although the non-specific RNA recognition by the N protein has been well-established (40, 41), this observation suggests that interactions with Ubl1 do not block the N protein from binding to this region of the viral genome.

## DISCUSSION

The urgent necessity to develop therapies against SARS-CoV-2 infections has focused research efforts on individual proteins encoded by this virus. The analysis of Spike, RdRp, and the two proteases, PlPro and 3Cpro, has been particularly intensive since these proteins represented the main targets of vaccination and antiviral therapies. These studies also highlighted the lack of extensive molecular knowledge about functional domains of Nsp3 protein beyond PlPro, although it represents the largest non-structural protein encoded in the SARS-CoV-2 genome. To bridge this gap, we pursued structural and functional analyses of multiple domains of this SARS-CoV-2 protein, providing the molecular activities and the first experimentally derived structures in the case of the βSM and a region of the Y domain, which have never been experimentally visualized until our structure determination.

We showed that multiple domains of Nsp3, including the Ubl1, ADRP, HSUD, NAB, and Y domains, recognize the 5′ UTR of the SARS-CoV-2 genome; this provides the first evidence of such activity in the cases of the ADRP and Y domains and the first indication of the molecular function of the Y domain. Our mutagenesis analysis highlighted the role of individual surface residues in the Ubl1 domain involved in interactions with viral RNA paving the way for further analysis of this activity. Our data pave the way for the identification of molecular determinants responsible for RNA binding in other Nsp3 domains that we have demonstrated to possess such activity. In the case of the ADRP domain, this activity will need to be reconciled with the enzymatic activity of this domain. Similarly, since the Y domain is expected to interact with Nsp4 and Nsp6, the effect of this interaction on RNA binding of this domain remains to be investigated. A recent cryogenic electron tomography (cryo-ET)-derived model of the pore complex (42) (PDB 8YAX) containing most of Nsp3 and Nsp4 showed that the Y domain is positioned in the cytoplasmic face of the pore. The domain is positioned next to the PlPro domain, with the Y4 region mediating this interaction. Interestingly, the Y4 region is not positioned within the central core of the pore, whereas each of the Y1–Y3 regions lines the central core; these observations are consistent with our data that the Y4 region does not interact with the 5′ UTR RNA, whereas the full Y domain possesses this functionality. In the cryo-ET structure of Nsp3-Nsp4, the HSUD and NAB domains are positioned distal to the core of the pore, and the model does not resolve the position of the Ubl1, ADRP, or βSM domains; how this positioning is consistent with our observation of RNA binding of the Ubl1, ADRP, and HSUD, NAB domains remains to be explained. Further structural studies of the pore, especially with RNA, are indeed warranted to complete our molecular and structural understanding of this complex structure.

Our analyses highlighted the role of the Ubl1 domain of Nsp3 as the key connector between this protein, the N protein, and the viral genome. Previous analysis using fluorescence polarization assay with short substrates (20 nucleotides) estimated the affinity of N protein to viral RNA to have the $K_D$ of ~7 nM (43). However, affinity decreased ~10-fold when the protein was incubated with SL RNA (43). This latter value is closer to the $K_D$ we obtained using EMSA assay against the 245 nucleotides of the 5′ UTR and this protein. We observed a 127-fold increase in affinity for the N protein in the presence of Ubl1 domain compared with N protein alone and 5,000-fold compared with calculated $K_D$ for Ubl1 alone. This clear result appears to conflict with the previously published observation (34) that the presence of Ubl1 decreases the affinity between the N protein and RNA. However, this may be due to differences between the experimental setup used in previously published work compared with ours. Specifically, our experiments were performed with a much larger fragment of RNA, and Ubl1 and N were preincubated to form a complex before their exposure to RNA rather than the formation of the N-RNA complex first. This comparative analysis suggests that the sequence of binding to RNA may have an impact on the affinity of the Ubl1-N complex compared with these proteins' interaction with RNA individually. One could envision as the RNA transits the DMV pore, the Ubl1 domain interacts with the 5′ UTR RNA first, and subsequently forms a complex with N protein, thus increasing the affinity for RNA. Such

mechanisms would be advantageous for packaging the newly transcribed viral genome. Additional experimental evidence of this model needs to be acquired.

According to our assay, the Ubl1 domain of Nsp3 shows specific binding to the first 245 bases of the 5′ UTR. In contrast, we observed no binding to the sequence immediately downstream from this region under the same experimental conditions. Although there may be multiple explanations, these results are supportive of Ubl1 possessing sequence-specific RNA binding activity. Ubl1 specificity to the 5′ UTR is consistent with this domain's established role in facilitating the binding of N protein to the first RNA bases exiting the DMV molecular pore and shielding it from cytoplasmic RNA sensors. Further studies are needed to delineate the specific sequence within the 5′ UTR recognized by Ubl1. Since the N-terminal fragment (residues 1–245) used in our assay includes predicted SL1–SL4 and a portion of SL5, the individual role of these secondary structure elements (39) in interactions with Ubl1 should be further clarified. Notably, the 5′ UTR contains three AUA sequences with one located in the 1–245 region, which were previously demonstrated to co-purify with SARS-CoV Ubl1, suggesting that these represent Ubl1 binding sites (20). Our mutagenesis and deletion analysis showed that the N-terminal β1 and β2 strands of Ubl1, as well as residues belonging to the α1 and α4 helices of this domain, are important for the recognition of the 5′ UTR. These results are consistent with the previously reported mutational analysis of the SARS-CoV Nsp3-Ubl1 complex, which demonstrated that alteration of residues belonging to the α1 helix affected interactions with ssRNA (20). Another yet unclarified aspect of the SARS-CoV-2 virus's life cycle is whether Ubl1 and the N-Ubl1 complex can discriminate between genomic RNA and subgenomic mRNA exiting the DMV pore. Presumably, the absence of the 5′ UTR on subgenomic mRNAs precludes their recognition by Ubl1, but further analysis is needed to clarify this.

During the preparation of this manuscript, a study was published that delineated the interaction region between Ubl1 and the N proteins and described the structure of this complex (36). The presented structure of the Ubl1-N complex contained only a single chain of Ubl1, whereas the sample used for structure reconstruction lacked the N-terminal portion (residues 1–15). A superposition of Ubl1 crystal structure with this domain in the complex with N protein showed that the conformation of N-termini (residues 1–14) does not introduce any steric clashes with the position of the N protein linker region, suggesting that dimerization of Ubl1 domain may be compatible with the formation of the complex with N protein. Observed interactions are also in line with the analysis of the Ubl1 equivalent domain called Nsp3a with N protein from the MHV virus, where this interaction was mapped to α2 of Nsp3a and the SR-rich region of N (35). The α2 helices of SARS-CoV-2 Ubl1 and MHV Nsp3a are similar in structure (Fig. S2) but show some primary sequence variation. Similarly, the N protein linker/serine-arginine (SR)-rich regions are similar between SARS-CoV-2 and MHV but do show variation in residue content. Specifically, the SARS-CoV-2 N protein contains one more arginine, two more serine residues, and one more hydrophobic residue in the hydrophobic region involved in interactions with Ubl1 according to our results.

The structure of the SARS-CoV2 Ubl1 domain is very similar to that of the corresponding fragments from SARS-CoV and MHV with the exception of the N-terminus. In both crystal forms obtained for SARS-CoV2 Ubl1, we observed the 14 N-terminal residues of this domain forming two β-strands involved in domain-swapping dimerization. We further demonstrate that the deletion of this N-terminal portion of Ubl1 abolishes both dimerization and the RNA-binding functionality of this domain. However, dimerization of Ubl1 may not be relevant *in vivo,* including when interacting with the N protein (34, 36) and in the context of the full DMV pore, as the cryo-ET structure of the Nsp3-Nsp4 pore suggests that the Ubl1 domains would be positioned quite far from each other.

We showed that the NAB of SARS-CoV-2 possesses affinity toward 5′-UTR ssRNA substrates. Previous characterization of NAB from SARS-CoV, which was not reported to oligomerize, was shown to bind to A- and G-rich RNAs, such as $(GGGA)_2$ and $(GGGA)_5$ (19). The 1–245 and the 301–545 base 5′-UTR ssRNA substrates used in this study do not

contain any GGG sequences, suggesting that NAB may recognize other RNA motifs that remain to be characterized.

The HSUD domain of Nsp3 from SARS-CoV and SARS-CoV-2 share 75% primary sequence identity. The HSUD of SARS-CoV-2 has been shown to bind G-quadruplexes/G4 sequences (18), and the HSUD from SARS-CoV has been shown to bind short RNA sequences generally rich in purines, as well as the TRS+ sequence in the 5′-UTR (29). Although three PQS (Potential G4 Sequences) are predicted for the 301–545 base region, there are no PQS predicted for the first 245 nucleotides of the 5′-UTR of the SARS-CoV-2 genome (44–47). Since we observed binding of the SARS-CoV-2 HSUD to both these fragments of 5′-UTR, this suggests that this domain's interactions with RNA may involve sequences beyond PQS.

To our knowledge, we are the first to report the RNA binding activity for the ADRP domain of SARS-CoV-2 Nsp3. This domain adopts a compact structure featuring charged surface patches that can be responsible for observed RNA binding (24, 27, 48). However, the role of individual ADRP residues in interactions with RNA and how this activity relates to the catalytic and protein interaction activities reported for this domain remains to be investigated.

The search for structurally similar proteins to the βSM domain did not reveal any significant hits, suggesting that this domain adopts a unique fold. Since the βSM domain lacked any affinity to tested fragments of 5′-UTR, the specific role of this domain remains unclear. Given its proximity to the transmembrane region of Nsp3, this suggests a potential role for the βSM domain in the orientation of the protein with respect to the DMV membrane and/or interactions with the membrane itself.

Overall, our results greatly expand the molecular data on individual domains of the largest protein encoded by the SARS-CoV-2 virus. According to the current model, SARS-CoV/CoV-2 viral genomes are shepherded onto the N protein by the Nsp3's Ubl1 domain, which interacts with both the RNA itself and the N protein. The crystal structures of the Ubl1 and βSM domains presented in this study have already been used to validate the models of multidomain fragments or full-length Nsp3 obtained by cryogenic electron tomography (cryo-ET) (11, 12).

## MATERIALS AND METHODS

### Cloning

The regions of SARS-CoV-2 *Orf1a* encoding the individual domains of Nsp3 were synthesized either by Twist Biosciences or using a BioXP 3200 (Codex DNA, San Diego, CA, USA) as codon-optimized for *E. coli* expression. As expressed as amino acids in mature Nsp3, domain boundaries of the individual domains were: Ubl1 1-111; ADRP 206-374; NAB 1,089–1,203 or 1,050–1,216; βSM 1,230–1,334; full Y 1,584–1,945; Y2–Y4 region 1,619–1,945; and Y4 region 1,844–1,945. The expression constructs for the N protein encoded all native 419 amino acids. Synthetic dsDNA was then cloned into the pMCSG53 expression vector. Note that purified HSUD and PlPro were provided as gifts from the labs of Karla Satchell and Andrjez Joachimiak, respectively.

### Protein expression and purification

Expression plasmids were transformed into *E. coli* BL21 Gold (DE3) (Stratagene, San Diego, CA, USA) cells harboring an extra plasmid encoding three rare tRNAs (AGG and AGA for Arg, ATA for Ile), and proteins were overexpressed in 1 L in ZYP-5052 auto-inducing complex medium (49) by incubating a few hours at 37°C, followed by transferring to 20°C for overnight growth. Cell pellets were collected by centrifugation at 6,000 × *g*. Ni-NTA affinity chromatography was used for protein purification. Cells were resuspended in binding buffer (50 mM HEPES pH 7.5, 500 mM NaCl, 5% glycerol [vol/vol]), 0.5 mM Tris(2-carboxyethyl) phosphine (TCEP), 5 mM MgCl$_2$, 1 mM phenylmethyl-sulfonyl fluoride (PMSF), and 1 mM benzamidine supplemented with 0.05% n-Dodecyl

β-D-maltoside (DDM) then lysed with a sonicator. After sonication and centrifugation (30 min at 20,000 × $g$ rpm; Avanti J-25 centrifuge, Beckman Coulter, Brea, California, USA), cleared lysates were applied to nickel-nitrilotriacetic acid (Ni-NTA) resin. Beads were washed, and proteins were eluted with loading buffer supplemented with 35 mM and 300 mM imidazole, respectively. Eluted His$_6$-RNA-binding (residues 47–173) and His$_6$-dimerization (residues 247–364) domains of the nucleocapsid protein were dialyzed against 0.3 M NaCl, 10 mM HEPES pH 7.5, 2.5 mM MgCl$_2$, 1 mM TCEP, and 1% (vol/vol) glycerol. His$_6$-N protein full-length alone and as a complex with His$_6$-Nsp3 Ubl1 domain (see SEC-RALS section below) and 5 mutants of Nsp3 Ubl1 (Δ1-14, D80RE81R, E26RE29R, E49RE52R, and K33EK38E) purified for RNA binding assay were further purified by size-exclusion chromatography on a Superdex 200 HiLoad 16/60 column equilibrated with buffer composed of 0.5 M NaCl, 5% (vol/vol) glycerol, 10 mM HEPES (pH7.5), 2 mM MgCl$_2$, and 10 mM β-mercaptoethanol. Other purified proteins were similarly purified by size exclusion chromatography on a Superdex 200 HiLoad 10/300 column equilibrated with the same buffer. Where necessary for crystallization or RNA binding, His$_6$ tags were cut by TEV protease (30 µg of TEV added to 1 mg of eluted protein) concurrently with dialysis at 4°C in either 300 mM NaCl, 10 mM HEPES pH 7.5, 1% (vol/vol) glycerol, 2.5 mM MgCl$_2$, 1 mM TCEP (for Ubl1 and N proteins), or 300 mM potassium chloride, 10 mM HEPES (pH 7.5) (for βSM) or 0.3 M potassium chloride, 0.5 mM TCEP, 2.5% (vol/vol) glycerol, and 1 mM MgCl$_2$ (for Y4 region). After dialyses, dialysisTEV mixtures were passed through 2nd Nickel-NTA to remove the His$_6$ tags, TEV, and uncut protein. All proteins were concentrated using a BioMax concentrator (EMD Millipore, Burlington, MA, USA), followed by passage through a 0.2 µm Ultrafree-MC centrifugal filtration device (EMD Millipore, Burlington, MA, USA) and stored at −80°C. Purity of proteins was checked using SDS-PAGE.

## Crystallization and x-ray structure determination

All crystals were grown at room temperature using the vapor diffusion sitting drop method using a Mosquito robot (SPT Labtech, Hertfordshire, UK). For Ubl1 form 1, 19 mg/mL protein was mixed with reservoir solution 1.6 M ammonium sulfate, 0.1 M HEPES pH 7.5, and 2% hexanediol, and the crystal was cryoprotected with reservoir solution plus 30% ethylene glycol. For Ubl1 form 2, 10 mg/mL of the Ubl1-N protein complex was mixed with reservoir solution 1.6 ammonium sulfate, 0.1 M HEPES pH 7.5, 2% hexanediol, and 1.25% 1-butyl-3-methylimidazolium dicynamide, and the crystal was cryoprotected with reservoir solution plus 25% ethylene glycol; note that only Ubl1 was found in the crystal. For NAB (residues 1,089–1,203), 15 mg/mL protein was mixed with reservoir solution 2 M ammonium sulfate and 2% hexanediol, and the crystal was cryoprotected with paratone oil. For βSM, 15 mg/mL protein was mixed with reservoir solution 0.5 M MES pH 6 and 40% tacsimate, and the crystal was cryoprotected with paratone oil. For the Y4 region, 8 mg/mL protein was mixed with reservoir solution 1.1 M sodium citrate and 0.1 M HEPES pH 7.5, and the crystal was cryoprotected with paratone oil. Diffraction data at 100 K were collected at a home source Rigaku Micro-max-007 rotating anode plus Rigaku R-AXIS IV detector, or, at beamline 19-ID of the Structural Biology Center at the Advanced Photon Source, Argonne National Laboratory. Diffraction data were processed using HKL3000 (50). Structures were solved by Molecular Replacement (MR) using Phenix.phaser (51) using the following models: for Ubl1, the Ubl1 domain from SARS-CoV (PDB 2GRI [20]); for NAB, the NAB domain from SARS-CoV (PDB 2K87 [19]); and for βSM and the Y4 region, models for MR were constructed by AlphaFold2 (38). Model building and refinement were performed using Phenix.refine and Coot (52). B-factors were refined as isotropic with TLS parameterization. Geometry was validated using Phenix.molprobity and the wwPDB validation server.

## Structural analysis

Oligomerization interfaces were analyzed using the PDBePISA server (53). Structural homologs in the PDB were searched for using the Dali-lite server (54) or the PDBeFold

server (55). Electrostatic solvent-accessible surfaces were calculated using PyMOL (Schrödinger, LLC, New York, NY, USA). Figures were created using PyMOL.

## SEC-RALS of Ubl1-N protein complex

To clarify its molecular weight and suggested stoichiometry, the $His_6$-Ubi1-N protein complex was produced by mixing individually purified $His_6$-Ubi1 and N proteins, followed by size exclusion chromatography on a Superdex 200 HiLoad 16/60 column equilibrated with buffer composed of 0.5 M NaCl, 5% (vol/vol) glycerol, 10 mM HEPES (pHpH 7), 2 mM $MgCl_2$, and 10 mM β-mercaptoethanol. Four peaks were observed in this chromatogram, with the second peak corresponding to the intact $His_6$-Ubi1-N protein complex as indicated by SDS-PAGE (Fig. S4a). Further molecular weight and shape analysis of this peak containing the $His_6$-Ubi1-N protein complex was carried out using size exclusion chromatography coupled with a 90° right-angle light scattering detector and 643 nm laser beam (OMNISEC Reveal, Malvern Panalytical, Malvern, UK). Before collecting any measurements, the protein was centrifuged at 10,000 × $g$ for 30 min at 4°C. The size exclusion analytical column (Bio-SEC-3, Agilent, Santa Clara, CA, USA) was loaded with 50 µL of protein at a concentration of 3.0 mg/mL. The protein was eluted through the column using a buffer composed of 250 mM NaCl, 20 mM HEPES pH 7.5, 5% glycerol, 5 mM $MgCl_2$, and 10 mM TCEP. Analysis of the data was performed using Malvern Analytical OMNISEC software. The molecular weight corresponded to 248,369, which approximately corresponds to a 4:4 complex. This same sample was used for EMSA analysis, see below.

## Preparation of nucleic acid substrates and electrophoretic mobility shift assay (EMSA)

The cDNA of SARS-CoV-2 was generated using the High Capacity cDNA Reverse Transcription Kit (Applied BioSystems, Waltham, MA, USA) from the MN908947.3 synthetic SARS-CoV-2 RNA (Twist Bioscience, South San Francisco, CA, USA). The DNA of the 5′-UTR region (1-245 bp) was amplified using PCR to include the T7 promoter with primers 5′-TAATACGACTCACTATAGGGATTAAAGGTTTATACCTTCC-3′ (forward) and 5′- GG ACGAAACCTAGATGTGCTGATGATCG-3′ (reverse). The DNA of the region downstream of 5′-UTR (301–545 bp) was amplified using PCR to include the T7 promoter with primers 5′-TAATACGACTCACTATAGGG ACACGTCCAACTCAGTTTG

−3′ (forward) and 5′- CTTCGAGTTCTGCTACCAGCTCAACCATAACATGAC −3′ (reverse).

Substrate ssRNA was transcribed using HiScribe T7 High Yield RNA Synthesis Kit (New England BioLabs, Ipswich, MA, USA) and was [$^{32}$P]-labeled at the 5′-end using T4 polynucleotide kinase (New England BioLabs) and purified as previously described (56). The reaction mixture for RNA binding assays with Ubl1, Ubl1 mutants, NAB, HSUD, and Y domains, as well as N protein and Ubl1-N protein complex, contained 50 mM Tris-HCl (pH 8), 150 mM NaCl, 5 mM $CaCl_2$, 1 mM DTT, 20 U RNaseOUT (Invitrogen, ThermoFisher, Waltham, MA, USA), and 8 nM (or 0.8 nM for reactions with the Nsp3-N protein complex) 5′-[$^{32}$P]-labeled RNA substrate. Reaction mixtures for RNA binding assays with the ADRP domain contained 50 mM Tris-HCl (pH 8), 150 mM NaCl, 10 mM $MgCl_2$, 1 mM DTT, 20 U RNaseOUT (Invitrogen), and 8 nM 5′-[$^{32}$P]-labeled RNA substrate. Note that all the proteins used in this assay had their His-tags removed except for the Ubl1 mutants. Reactions were incubated for 1 h at 37°C, quenched by the addition of glycerol loading dye, and separated on 6% native polyacrylamide gels. The results were visualized using a Phosphoimager, with the percentage of bound substrate quantified using ImageLab software (Bio-Rad, Hercules, CA, USA). Values were plotted against total protein concentration to determine $K_D$ values using non-linear regression fit in Prism software (GraphPad, San Diego, CA). Replicate EMSA gels for Fig. 2 and 3 are shown in Fig. S5.

## ACKNOWLEDGMENTS

We thank Rosa Di Leo and Cameron Semper for cloning. We thank Robert Flick and the BioZone Mass Spectrometry facility for validation of protein purity. We thank Monica Rosas Lemus in Karla Satchell's laboratory for purified HSUD protein, and Christine Tenar and Jurek Osipiuk in Andrzej Joachimiak's laboratory for purified PlPro protein. We thank Changsoo Chang and Youngchang Kim at the Structural Biology Center, Advanced Photon Source, Argonne National Laboratory for x-ray diffraction data collection. The Advanced Photon Source, a U.S. Department of Energy (DOE) Office of Science User Facility operated for the DOE Office of Science by Argonne National Laboratory, and the Structural Biology Center beamline resources are supported under contract no. DE-AC02-06CH11357. This project was funded by the University of Toronto COVID-19 Action Initiative awarded to Aled Edwards. This project has also been funded in part with U.S. Federal funds from the National Institute of Allergy and Infectious Diseases, National Institutes of Health (NIH), Department of Health and Human Services, under Contract numbers HHSN272201700060C and 75N93022C00035.

## AUTHOR AFFILIATIONS

[1]BioZone, Department of Chemical Engineering and Applied Chemistry, University of Toronto, Toronto, Canada
[2]Department of Microbiology, Immunology and Infectious Diseases, University of Calgary, Calgary, Canada
[3]Center for Structural Biology of Infectious Diseases (CSBID), Calgary, Alberta, Canada

## AUTHOR ORCIDs

Sofia Lemak  http://orcid.org/0009-0009-3479-1609
Peter J. Stogios  http://orcid.org/0000-0001-8663-1425
Alexei Savchenko  http://orcid.org/0000-0002-5256-9237

## FUNDING

| Funder | Grant(s) | Author(s) |
| --- | --- | --- |
| National Institute of Allergy and Infectious Diseases | HHSN272201700060C | Alexei Savchenko |
| National Institute of Allergy and Infectious Diseases | 75N93022C00035 | Alexei Savchenko |
| Temerty Faculty of Medicine, University of Toronto | University of Toronto COVID-19 Action Initiative | Peter J. Stogios |

## AUTHOR CONTRIBUTIONS

Sofia Lemak, Conceptualization, Data curation, Formal analysis, Funding acquisition, Investigation, Methodology, Project administration, Resources, Software, Supervision, Validation, Visualization, Writing – original draft, Writing – review and editing | Tatiana Skarina, Data curation, Formal analysis, Funding acquisition, Investigation, Methodology, Project administration, Resources, Supervision, Visualization, Writing – original draft, Writing – review and editing | Deepak T. Patel, Data curation, Formal analysis, Investigation, Methodology, Visualization, Writing – original draft, Writing – review and editing | Peter J. Stogios, Conceptualization, Data curation, Formal analysis, Funding acquisition, Investigation, Methodology, Project administration, Resources, Software, Supervision, Validation, Visualization, Writing – original draft, Writing – review and editing | Alexei Savchenko, Data curation, Formal analysis, Funding acquisition, Project administration, Resources, Supervision, Visualization, Writing – review and editing

## DATA AVAILABILITY

All crystal structures are available in the Protein Databank under accession codes 7KAG, 7TI9, 7LGO, 7T9W and 7RQG. SDS-PAGE and EMSA gel images shown represent the original images and as such are available in this manuscript. SEC and SEC-RALS data can be found at Figshare: https://doi.org/10.6084/m9.figshare.25800580

## ADDITIONAL FILES

The following material is available online.

### Supplemental Material

**Supplemental figures and tables (Spectrum02871-24-S0001.pdf).** Fig. S1 to S6 and Table S1.

### Open Peer Review

**PEER REVIEW HISTORY (review-history.pdf).** An accounting of the reviewer comments and feedback.

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
