## [Reviewer comments · Microbiology Spectrum]

Microbiology Spectrum

Structural and functional analyses of SARS-CoV-2 Nsp3 and its specific interactions with the 5' UTR of the viral genome

Sofia Lemak, Tatiana Skarina, Deepak Patel, Peter Stogios, and Alexei Savchenko

Corresponding Author(s): Peter Stogios, University of Toronto

Review Timeline:

Submission Date:	November 25, 2024
Editorial Decision:	February 22, 2025
Revision Received:	April 14, 2025
Accepted:	June 4, 2025

Editor: Vaithilingaraja Arumugaswami

Reviewer(s): The reviewers have opted to remain anonymous.

Transaction Report:

DOI: <https://doi.org/10.1128/spectrum.02871-24>

Re: Spectrum02871-24 (Structural and functional analyses of SARS-CoV-2 Nsp3 and its specific interactions with the 5' UTR of the viral genome)

Dear Dr. Peter J. Stogios:

Thank you for the privilege of reviewing your work. Below you will find my comments, instructions from the Spectrum editorial office, and the reviewer comments.

We made this decision by considering both the responses to previous critiques and the new feedback received. Please return the manuscript within 60 days; if you cannot complete the modification within this time period, please contact me. If you do not wish to modify the manuscript and prefer to submit it to another journal, notify me immediately so that the manuscript may be formally withdrawn from consideration by Spectrum.

Revision Guidelines

Sincerely,
Vaithilingaraja Arumugaswami
Editor
Microbiology Spectrum

Reviewer #1 (Comments for the Author):

Summary & Significance

Lemak et al. report crystallographic and biochemical analyses of the Nsp3 protein from SARS-CoV-2. This protein is an essential

component of double-membrane vesicles that function as RNA processing centers during viral replication. Nsp3 is suspected to form a transmembrane pore that exports genomic RNA from the vesicle and initiates packaging with the Nucleocapsid (N) protein. The authors investigate individual domains of Nsp3 to dissect their function and succeed in determining the structures of four regions. For the N-terminal Ubl1 domain, they make the novel finding that the first 14 residues adopt a β -hairpin fold that mediates dimerization of the domain, a feature not previously observed in homologous crystal structures. They also report the first structure of the uncharacterized β SM domain, which has a unique fold that will hopefully be connected to the role of this segment in future studies. Furthermore, the authors show that multiple domains of Nsp3 can independently interact with the 5'-UTR of the viral genome, with varying degrees of affinity and specificity. Perhaps the most striking finding here is that a complex of the N protein with the Ubl1 domain strongly increases binding affinity, suggesting Nsp3 and N may cooperate to discriminate viral genomic RNA from sub-genomic RNA or other competitors via multivalency. Although several similar structures have been recently reported by other groups, this study still offers significant new insights.

Major Critiques

1. For RNA binding studies with the Ubl1 domain, it is difficult to determine where recombinant protein has been used with the His-tag still attached, and where the tag has been removed by TEV protease. For example, Supplemental Data Figure 1a shows an SDS-PAGE gel of WT Ubl1 with the His-tag removed alongside the mutants which retain the His-tag. Did the WT protein used in the binding assays also retain the tag? Given that the authors argue that the N-terminal region of this domain mediates dimerization and is necessary for RNA binding, they should show that appending a His-tag at the N-terminus does not interfere with these functions. One way to address this would be to include the SEC data for the Ubl1 mutants mentioned in main text line 253.

Minor Critiques

1. In Supplemental Data Figure 1, the authors may provide the UV absorption spectrum of the proteins to demonstrate that they have purified away any contaminating RNA from the E. coli host.
2. Supplemental Data Figure 1 does not include SDS-PAGE gels for the HSUD, ADRP, or N protein samples used in this study.
3. In Supplemental Data Figure 1b-h, SEC experiments are performed with an analytical column, except for panel 1c, where the retention volume is consistent with a preparative scale column. Perhaps the authors can indicate which column is used for each experiment in the figure or elaborate in the Methods section.
4. In Supplemental Figure 1d, the SEC retention volume of the NAB (1089-1203) construct is interpreted as indicating a molecular weight of 14.3 kDa, or a monomeric specie. However, the peak clearly elutes before the 17 kDa marker protein and is closer to the 44 kDa marker. This would indicate the molecular weight to be substantially larger than the monomer. The same issue applies to construct Y4 (1844-1945) in panel 1h.
5. In Supplemental Data Table 1, for the Nsp3 NAB (7LGO) crystal, the resolution limits reported under the Data Collection section are not consistent with the limits given in the Refinement section (that is, 2.44 Å data was collected, but the model was refined against 1.74 Å data).
6. In Supplemental Data Table 1, for crystals Nsp3 β SM (7T9W) and Nsp3 Y4 (7RQG), the reported number of unique reflections does not seem consistent. Since both crystals diffract to about the same resolution (~2.2 Å), we would expect to see fewer unique reflections for β SM, since this crystallizes in the much higher symmetry space group P43. Perhaps these numbers represent the total, instead of unique, reflections observed?
7. In the main text, lines 157-158, please report the α -RMSD between the form 1 and form 2 structures.
8. In the main text, line 185, the authors may wish to calculate the buried surface area of the interface between the two chains to help gauge if this is a "crystallographic dimer" or is relevant to the dimeric solution structure suggested by the NAB 1050-1216 fragment.
9. In the main text, line 194, can the difference in conformation of these linkers be attributed to differing lattice interactions in the two structures?
10. The SEC-RALS experiment shown in Supplemental Data Figure 4 is described in the Materials Methods section but not elaborated on in the Results or Discussion. Does this higher order complex have some biological significance?
11. Some of the gel shift assays (for example Figure 3C) reveal activity retained in the pocket of the gel, which is not uncommon for RNA-binding proteins. However, it does complicate the quantification, as these complexes are typically not included in the estimation of the fraction bound. The authors may consider to indicate only a rough estimate of the affinity for these cases.

Errata

1. In main text line 167, the reference to the SEC data should point to Supplemental Data Fig. 1c, not 1b.
2. In main text line 181, "NAR" should be "NAB".
3. In main text line 225, the figure reference should point to Supplemental Data Fig. 2d and 2e.

Summary & Significance

Lemak et al. report crystallographic and biochemical analyses of the Nsp3 protein from SARS-CoV-2. This protein is an essential component of double-membrane vesicles that function as RNA processing centers during viral replication. Nsp3 is suspected to form a transmembrane pore that exports genomic RNA from the vesicle and initiates packaging with the Nucleocapsid (N) protein. The authors investigate individual domains of Nsp3 to dissect their function and succeed in determining the structures of four regions. For the N-terminal Ubl1 domain, they make the novel finding that the first 14 residues adopt a β -hairpin fold that mediates dimerization of the domain, a feature not previously observed in homologous crystal structures. They also report the first structure of the uncharacterized β SM domain, which has a unique fold that will hopefully be connected to the role of this segment in future studies. Furthermore, the authors show that multiple domains of Nsp3 can independently interact with the 5'-UTR of the viral genome, with varying degrees of affinity and specificity. Perhaps the most striking finding here is that a complex of the N protein with the Ubl1 domain strongly increases binding affinity, suggesting Nsp3 and N may cooperate to discriminate viral genomic RNA from sub-genomic RNA or other competitors via multivalency. Although several similar structures have been recently reported by other groups, this study still offers significant new insights.

Major Critiques

1. For RNA binding studies with the Ubl1 domain, it is difficult to determine where recombinant protein has been used with the His-tag still attached, and where the tag has been removed by TEV protease. For example, Supplemental Data Figure 1a shows an SDS-PAGE gel of WT Ubl1 with the His-tag removed alongside the mutants which retain the His-tag. Did the WT protein used in the binding assays also retain the tag? Given that the authors argue that the N-terminal region of this domain mediates dimerization and is necessary for RNA binding, they should show that appending a His-tag at the N-terminus does not interfere with these functions. One way to address this would be to include the SEC data for the Ubl1 mutants mentioned in main text line 253.

Minor Critiques

1. In Supplemental Data Figure 1, the authors may provide the UV absorption spectrum of the proteins to demonstrate that they have purified away any contaminating RNA from the *E. coli* host.
2. Supplemental Data Figure 1 does not include SDS-PAGE gels for the HSUD, ADRP, or N protein samples used in this study.
3. In Supplemental Data Figure 1b-h, SEC experiments are performed with an analytical column, except for panel 1c, where the retention volume is consistent with a preparative scale column. Perhaps the authors can indicate which column is used for each experiment in the figure or elaborate in the Methods section.
4. In Supplemental Figure 1d, the SEC retention volume of the NAB (1089-1203) construct is interpreted as indicating a molecular weight of 14.3 kDa, or a monomeric specie. However, the peak clearly elutes before the 17 kDa marker protein and is closer to the 44 kDa marker. This would indicate the molecular weight to be substantially larger than the monomer. The same issue applies to construct Y4 (1844-1945) in panel 1h.

5. In Supplemental Data Table 1, for the Nsp3 NAB (7LGO) crystal, the resolution limits reported under the *Data Collection* section are not consistent with the limits given in the *Refinement* section (that is, 2.44 Å data was collected, but the model was refined against 1.74 Å data).
6. In Supplemental Data Table 1, for crystals Nsp3 βSM (7T9W) and Nsp3 Y4 (7RQG), the reported number of unique reflections does not seem consistent. Since both crystals diffract to about the same resolution (~2.2 Å), we would expect to see *fewer* unique reflections for βSM, since this crystallizes in the much higher symmetry space group P4₃. Perhaps these numbers represent the total, instead of unique, reflections observed?
7. In the main text, lines 157-158, please report the Cα-RMSD between the form 1 and form 2 structures.
8. In the main text, line 185, the authors may wish to calculate the buried surface area of the interface between the two chains to help gauge if this is a “crystallographic dimer” or is relevant to the dimeric solution structure suggested by the NAB 1050-1216 fragment.
9. In the main text, line 194, can the difference in conformation of these linkers be attributed to differing lattice interactions in the two structures?
10. The SEC-RALS experiment shown in Supplemental Data Figure 4 is described in the Materials Methods section but not elaborated on in the Results or Discussion. Does this higher order complex have some biological significance?
11. Some of the gel shift assays (for example Figure 3C) reveal activity retained in the pocket of the gel, which is not uncommon for RNA-binding proteins. However, it does complicate the quantification, as these complexes are typically not included in the estimation of the fraction bound. The authors may consider to indicate only a rough estimate of the affinity for these cases.

Errata

1. In main text line 167, the reference to the SEC data should point to Supplemental Data Fig. 1c, not 1b.
2. In main text line 181, “NAR” should be “NAB”.
3. In main text line 225, the figure reference should point to Supplemental Data Fig. 2d and 2e.

Response to mBio reviews

We appreciate the time and effort by the reviewers of our manuscript submitted to mBio and are thankful for the opportunity to address them and improve our paper. The original reviewers' comments are enumerated below, and our point-by-point replies are in *italics*.

We are confident we have addressed essentially all of their concerns.

Reviewer #1's comments:

While the work is important, some of the results and claims should be supported with more data and better described.

We thank the reviewer for the comment and addressed your comment regarding better describing data in the "major concerns" points below.

Finally, figure legends for main and supplementary figures were not provided, which is unacceptable.

We apologize for this oversight. We have added all necessary figure legends for main and supplemental data figures.

Major concerns:

1) No figure legends were provided and thus there is missing information on the number of repetitions and the experimental setup.

We apologize for this oversight. We have added all necessary figure legends for main and supplemental data figures.

2) Please provide how many times was the electrophoretic mobility shift assay repeated and if that was done always with the same protein purification batch or if different preparations were used. The plot in Figure 2D, and many points in 3C and 3D do not show error bars. Was this done only once? If yes, this should be repeated.

The EMSA's were repeated three times with the same protein purification batch. The figure legends have been updated accordingly and we have added a new supplemental figure (Supplemental Data Figure 5) to show two replicate EMSAs. All figures have been corrected to add error bars.

3) Data quality: Figure 3B: the electrophoresis does not show proper bands. Why is this and could this be improved? Supplementary Figure 3B is noisy.

The EMSA was carried out using RNA prepared experimentally using PCR amplification of SARS-CoV-2 cDNA (also prepared in the lab, (see Methods) using primers designed to produce a fragment of the first 245nt of the genome. In addition, the ³²P-labelling of RNA relies on a gel purification step in which the labelled RNA of the correct size is excised for further purification. These steps can produce an RNA substrate that is a mixture of 245nt RNA and lower size RNA (due to PCR error, degradation, etc), and since the native gel assay resolves nucleotides with even small size differences, all labelled RNA species are present in the blank substrate lane. Since the RNA had to be labelled several times throughout the study due to the short half-life of ³²P, substrates from different experiments can look different as a result of the gel purification step. In our manuscript we did not over-analyze the shape of the bands but simply focused on their presence and disappearance which is indicative of RNA binding.

Supplemental Figure 3B is updated with an image of the same gel that we attempted to “de-noise”.

4) Authors claim that the binding is specific to RNA 1-301 but as data supporting this conclusion, they provide only one assay performed using RNA 301-545 in Supplementary Fig 3 C and neither binding curves nor concentrations/ratio RNA/Proteins are provided. This should be provided and properly described.

The added Supplemental Data Figure Legends now indicates the concentration of proteins used (the Materials and Methods indicates the concentration of RNA used, which is 8 nM). We do not see any value in adding binding curves where no RNA binding was observed to this RNA by the Ubl1, NAB, Y and Ubl1(delta1-14) domains.

5) The manuscript would be improved by showing a schematic of the loops present in 5'UTR and the rationale for selecting 1-301 and 301-545. 1-301 contains a large complex loop SL5 between 140-300 which is composed of other loops SL5A, B and C. Authors say that 1-301 RNA contains only a portion of SL5. Hence it is difficult to know if SL5 is a determinant of specificity.

We appreciate this comment and certainly agree that a more rigorous dissection of the secondary structure of the 5' UTR as it relates to RNA binding Nsp3 is important. However this was outside of the scope of our funding and personnel resources. We updated the Discussion to emphasize the importance of this research:

“Further studies are needed to delineate the specific sequence within the 5' UTR recognised by Ubl1. Since the N-terminal fragment 348 (residues 1-245) used in our

assay includes predicted SL1 through SL4, and a portion of SL5, individual role of these secondary structure elements in interactions with Ubl1 should be further clarified.”

6) Previously it has been shown that SARS-CoV-2 Ubl1 competes with RNA to bind N protein in a dose-dependent manner (doi.org/10.1038/s42003-023-04570-2). This differs from the authors' suggestion that the binding to RNA binding for the Ubl1-N complex is synergistic. What is added first in the binding assay, N or RNA to the Ubl1? How does that change if it is performed in another sequence?

We appreciate the reviewer identifying this apparent discrepancy, and we agree that the order of binding may be an important factor behind this. There were numerous differences between the referenced paper and our study. Firstly, we used a much longer fragment of RNA than the one used by the authors of the referenced paper; the longer fragment of RNA may present additional binding sites for N, Ubl1 or N-Ubl1 that are not present on the shorter fragment, which would confound comparison of the results. Secondly, the authors of paper referenced undertook experiments where the N protein – RNA complex were formed first, followed by addition of Ubl1 and then observed an increase in the concentration of free RNA, suggesting that Ubl1 bound to N protein and excluded RNA. In our study, we formed the N-Ubl1 complex first, followed by addition of RNA, and we still observed RNA binding. A comparison of these two studies would require matching RNA and order of protein addition. Regardless, we agree that the sequence of formation of the N-Ubl1 complex, vs the N-Ubl1-RNA, N-RNA, or Ubl1-RNA complexes is important and should be dissected further. We updated the Discussion section with these thoughts:

“This clear result appears to conflict with the previously published observation (34) that the presence of Ubl1 decreases the affinity between the N protein and RNA. However, this may be due to differences between experimental set up used in previously published work compared to ours. Specifically, our experiments were performed with a much larger fragment of RNA, and Ubl1 and N were preincubated to form a complex before their exposure to RNA rather than formation of the N- RNA complex first. This comparative analysis suggests that the sequence of binding to RNA may have an impact on the affinity of the Ubl1-N complex as compared to these proteins' interaction with RNA individually. One could envision as the RNA transits the DMV pore, the Ubl1 domain interacts with the 5' UTR RNA first, and subsequently forms a complex with N protein thus increasing the affinity for RNA. Such mechanism would be advantageous for packaging the newly transcribed viral genome. Additional experimental evidence of this model needs to be acquired.”

7) Ubl1 and Y are positioned on opposite sides of the pore. Ubl1 is positioned outwards and Y domain is placed inside the double membrane vesicle, which contains a double-stranded RNA intermediate of RNA synthesis. This could be discussed. In addition, it is plausible that Y specifically binds another part of the RNA genome, hence providing data on whether double-stranded RNA binds to Y would provide further insight.

We absolutely agree that the relative positioning of the Ubl1 and Y domains on the pore would play a role in the process of binding RNA as it exists the pore. In the process of preparing the revised version of our study, a cryo-electron tomography model of the pore complex containing most of Nsp3 and Nsp4 was released (<https://doi.org/10.1038/s41586-024-07817-y>, our reference 42). The Y1-Y4 region is visible on the cytoplasmic side of the pore. The Ubl1 domain is not visible, but would have to also be located on the cytoplasmic side according to the location of the transmembrane region of Nsp3. We updated the Discussion section (lines 307-319) in light of these findings but also indicate that further structural studies are necessary to truly understand the relative positioning of all regions of Nsp3 and the RNA.

“Interestingly, a recent cryogenic electron tomography (cryo-ET)-derived model of the pore complex (42) (PDB 8YAX) containing most of Nsp3 and Nsp4 showed that the Y domain is positioned in the cytoplasmic face of the pore. The domain is positioned next to the PIP₂ domain, with the Y4 region mediating this interaction. Interestingly, the Y4 region is not positioned within the central core of the pore, while each of the Y1 through Y3 regions line the central core; these observations are consistent with our data that the Y4 region does not interact with the 5' UTR RNA, while the full Y domain possesses this functionality. In the cryo-ET structure of Nsp3-Nsp4, the HSUD and NAB domains are positioned distal to the core of the pore, and the model does not resolve the position of the Ubl1, ADRP or β SM domains; how this positioning is consistent with our observation of RNA binding of the Ubl1, ADRP and HSUD, NAB domains remains to be explained. Further structural studies of the pore, especially with RNA, are indeed warranted to complete our molecular and structural understanding of this complex structure.”

Regarding potential binding of Y to another part of the RNA genome; this is certainly an intriguing possibility. Unfortunately, due to funding and personnel constraints, research on this was outside of the scope of this study.

8) Although it is possible that purified Ubl1 domain can form oligomers in vitro, it is difficult to imagine how this would happen in infected cells. The crown structure of nsp3-4 does fit 6 nsp3 proteins with Ubl1 positioned at the extreme N-terminus. It is not clear how would Ubl1 be able to oligomerize at this position since they are far apart. Please comment on this.

We certainly acknowledge this comment and agree that in cellulo, dimerization of Ubl1 may not be possible. We modified the Discussion section accordingly:

Deleted: “Based on this analysis we hypothesised that the RNA binding surface and N-protein interaction surfaces of Ubl1 are fully formed only upon its dimerization. Further studies are necessary to define the role of Ubl1 dimerization in the context of full-length Nsp3 and its role in the DMV molecular pore.”

Added: "However, dimerization of Ubl1 may not be relevant in vivo, including when interacting with the N protein (34, 36) and in the context of the full DMV pore, as the cryo-ET structure of the Nsp3-Nsp4 pore suggests that the Ubl1 domains would be positioned quite far from each other."

Minor comments:

1) It would be interesting to use AlphaFold 3 server to predict the structure of Ubl1-RNA and Ubl1-N-RNA and mutate the predicted sites in Ubl1. Finally, this could be compared with the structure of the N and Ubl1 complex that was published (Ni X et al)

This was a great suggestion. We attempted to predict the structure of Ubl1-RNA and Ubl1-N-RNA with AlphaFold3 but with little success. We used multiple different lengths of RNA comprising portions of the 5' UTR (bases 1 to 245) but the calculated models did not position the RNA interacting with the protein(s) to any significant degree in terms of polar, H-bond interactions and buried surface area. Since we could not generate such a model, we could not perform the suggested comparison to the published N-Ubl1 complex structure.

2) Consider rephrasing this sentence to improve the clarity: "In line with significant sequence conservation between the SARS-CoV-2 Nsp3 Ubl1 domain with this domain in Nsp3 from SARS-CoV and MHV viruses (sharing 75% and 31% of sequence identity, correspondingly) the structures of these domains superimposed with RMSD of 2.7Å and 4.1 Å, over 102 and 100 C α atoms, respectively vs Ubl1 from SARS-CoV-2, Supplemental Data Fig.2b)."

We reworded this sentence to:

"In line with significant sequence conservation between the SARS-CoV-2 Nsp3 Ubl1 domain with the Nsp3 Ubl1 domains from SARS-CoV and MHV (sharing 75% and 31% sequence identity, correspondingly), these structures superimposed with RMSD of 2.7Å and 4.1 Å, over 102 and 100 C α atoms, respectively (Supplemental Data Fig. 2b)."

3) 336, page 15: typo: fromo

Corrected.

Reviewer #2:

The manuscript by Lemak et al. presents an extensive structure-function analysis of the

SARS-CoV-2 Nsp3 protein, including the determination of crystal structures for several domains. However, this work does not offer significant new insights. Most of the presented structures, including the Ubl1, NAB, β SM domains, and a sub-region of the Y domain, have already been published by other labs, even when the authors for this manuscript deposited the PDBs years ago. This reduces the novelty and impact of the current study, rendering it more of an afterthought than an original contribution.

We respectfully disagree with the comment that “most of the presented structures .. have been published by other labs”, actually our structure of the NAB and β SM domains are the only structures of these domains from SARS-CoV-2 that are published.

The investigation into the RNA binding capacity of these domains adds little to the existing body of knowledge, as these interactions have been well-documented previously.

We disagree that the RNA binding capacity of these domains “adds little to the existing body of knowledge as these interactions have been well-documented previously”. These interactions have not been studied with the 5’ UTR from SARS-CoV-2, and there has been no RNA binding function known for the HSUD, ADRP and Y domains; in fact, there has been no function at all known for the Y domain.

Moreover, the manuscript's main conclusion, that the Ubl1-N protein complex binds the 5' UTR of the viral genome with greater affinity than the individual proteins, contradicts established models in the literature.

Regarding our finding that the Ubl1-N protein complex binds the 5’ UTR with greater affinity; we stand by our experimental results, but we also improved the paper by describing the differences in the experimental set up done by other studying this interaction (notably, <https://doi.org/10.1038/s42003-023-04570-2>). The Discussion was updated accordingly:

“This clear result appears to conflict with the previously published observation (34) that the presence of Ubl1 decreases the affinity between the N protein and RNA. However, this may be due to differences between experimental set up used in previously published work compared to ours. Specifically, our experiments were performed with a much larger fragment of RNA, and Ubl1 and N were preincubated to form a complex before their exposure to RNA rather than formation of the N- RNA complex first. This comparative analysis suggests that the sequence of binding to RNA may have an impact on the affinity of the Ubl1-N complex as compared to these proteins’ interaction with RNA individually. One could envision as the RNA transits the DMV pore, the Ubl1 domain interacts with the 5’ UTR RNA first, and subsequently forms a complex with N protein thus increasing the affinity for RNA. Such mechanism would be advantageous for packaging the newly transcribed viral genome. Additional experimental evidence of this model needs to be acquired.”

The supporting data, particularly the EMSA RNA gel shift assays, are of poor quality, raising doubts about the reliability of the results.

We updated the paper to show multiple replicates of the gels (Supplemental Data Figure 5), increased the quality of the gel images (particularly Figure 2d and Supplemental Figure 3B).

The authors fail to acknowledge or discuss relevant published data that conflict with their findings, undermining the manuscript's scientific rigor.

We updated the paper to discuss relevant published data, including the section indicated in the point above, as well as discussion of a cryo-electron tomography structure of Nsp3-Nsp4 (<https://doi.org/10.1038/s41586-024-07817-y>).

The manuscript is poorly written, with numerous issues such as missing figure legends and construct information, and the absence of a crucial crystallographic data table. These shortcomings alone justify rejection, as they indicate a lack of thoroughness and attention to detail. In conclusion, due to the lack of novel information, the questionable quality of the data, and the substandard presentation, in this reviewer's opinion, this manuscript does not meet the necessary standards for publication in mBio.

We apologize for the oversight in missing figure legends, construct information and crystallographic stats table. These have been added to the revised version of this paper.

Some of the other major issues:

1. Introduction length and outdated information. The introduction is excessively long yet still outdated, missing recent literature. It is common knowledge that Nsp3 has at least 16 known domains, not seven as mentioned in the manuscript. A recent paper on CoV structure (not the full-length Y as mentioned in the manuscript) has established that the Y domain of Nsp3 consists of four subdomains (Y1-Y4). However, the manuscript still presents incorrect domain boundaries in the figure and incorrectly named the subdomain they determined as Y3, which should be Y4.

We updated the Introduction to discuss the Y1-Y4 domains:

“The Y domain has been subdivided into four regions: Y1, Y2, Y3 and Y4 (37), with Y2 and Y3 regions restricted to Coronaviridae (32) and the Y2 and Y4 regions sharing higher conservation.”

And we also updated Figure 1 to show the Y1 through Y4 domains and their boundaries.

2. Ubl1 dimerization. The authors provide an extensive description to argue that Ubl1

forms a dimer. However, previous publications have already shown that the stoichiometry of N and Ubl1 binding is 1:2, supported by the crystal structure of the complex. Instead, the manuscript should compare the conformation of the Ubl1 dimer when alone and in the N-Ubl1 complex.

We agree that dimerization may not be relevant in vivo and we added a sentence in the Discussion section accordingly:

“However, dimerization of Ubl1 may not be relevant in vivo, including when interacting with the N protein (34, 36) and in the context of the full DMV pore, as the cryo-ET structure of the Nsp3-Nsp4 pore suggests that the Ubl1 domains would be positioned quite far from each other.”

We also point out to the reviewer that we did compare our Ubl1 dimeric structure with Ubl1 in the Ubl1-N crystal structure:

“During preparation of this manuscript, a study was published that delineated the interaction region between Ubl1 and the N protein and described the structure of this complex (36). The presented structure of the Ubl1-N complex contained only a single chain of Ubl1, while the sample used for structure reconstruction lacked the N-terminal portion (residues 1-15). A superposition of Ubl1 crystal structure with this domain in the complex with N protein showed that the conformation of N-termini (residues 1-14) does not introduce any steric clashes with the position of the N protein linker region, suggesting that dimerization of Ubl1 domain may be compatible with formation of complex with N protein.

3. EMSA assay quality. The EMSA assay is problematic. It is unclear why the authors chose a 5'-UTR RNA with a length of 245bp. The gel shift images are poor, showing multiple bands even for the RNA itself (presumably the control, though there is no figure legends). It is unclear how the authors defined the shifted bands or calculated Kd values to two decimal places. In addition, in some EMSA assays, the same 245bp RNA bands suddenly appear very sharp for Ubl1-N, HSUD and Y.

The EMSA was carried out using RNA prepared experimentally using PCR amplification of SARS-CoV-2 cDNA (also prepared in the lab, (see Methods) using primers designed to produce a fragment of the first 245nt of the genome. In addition, the ³²P-labelling of RNA relies on a gel purification step in which the labelled RNA of the correct size is excised for further purification. These steps can produce an RNA substrate that is a mixture of 245nt RNA and lower size RNA (due to PCR error, degradation, etc), and since the native gel assay resolves nucleotides with even small size differences, all labelled RNA species are present in the blank substrate lane. Since the RNA had to be labelled several times throughout the study due to the short half-life of ³²P, substrates from different experiments can look different as a result of the gel purification step.

We have included replicates of the EMSAs as a new Supplemental Data Figure 5 which confirms that the gel results are reproducible.

Regarding how we defined the shifted bands and calculated K_d values: The control (first lane) band(s) in each gel was considered to be 100% substrate, and the % substrate remaining in the other lanes was calculated using ImageLab software - we updated Materials and Methods accordingly:

The control (first lane) band(s) in each gel was considered to be 100% substrate, and the % substrate remaining in the other lanes was calculated using ImageLab.

K_d values were calculated using GraphPad software based on those percentages. We indicate that the K_d values are approximate, given the variability introduced by the experimental procedure, and experiments were done in triplicates to rule out random error, therefore we are confident in our methodology and conclusions.

4. Method. There is no description of the N protein construct used in the EMSA assay, how it was purified or how the N-Ubl1 complex was made. There is no crystallographic table that is a standard for listing the information for all the structures.

We added the construct information to the Materials and Methods – Cloning section. Information on purification of N-Ubl1 complex already appears in the Materials and Methods – SEC-RALS section. We added the crystallographic statistics table as Supplemental Data Table 1.

Response to Reviewers

Reviewer #1

Major critiques:

1. For RNA binding studies with the Ubl1 domain, it is difficult to determine where recombinant protein has been used with the His-tag still attached, and where the tag has been removed by TEV protease. For example, Supplemental Data Figure 1a shows an SDS-PAGE gel of WT Ubl1 with the His-tag removed alongside the mutants which retain the His-tag. Did the WT protein used in the binding assays also retain the tag? Given that the authors argue that the N-terminal region of this domain mediates dimerization and is necessary for RNA binding, they should show that appending a His-tag at the N-terminus does not interfere with these functions. One way to address this would be to include the SEC data for the Ubl1 mutants mentioned in main text line 253.

We apologize for the lack of clarity with respect to the tag on the Ubl1 domain used in RNA binding studies. The tag was removed. We have updated the Materials and Methods with the following sentences to clarify:

Line 468: Where necessary for crystallization or RNA binding, His₆ tags were cut off

Line 548: Note that all the proteins used in this assay had their His-tags removed except for the Ubl1 mutants.

Regarding the effect of the His-tag on dimerization, we added a new Supplemental Data Figure 6 showing the SEC profiles for two selected Ubl1 variants, carrying the His₆ tag. These profiles suggested that both selected Ubl1 variants are predominantly dimeric, consistent with the His₆ tag not affecting dimerization. We updated the main text (lines 264) to change “data not shown” to:

(Supplemental Data Figure 6 for two representative variants).

Minor critiques:

1. In Supplemental Data Figure 1, the authors may provide the UV absorption spectrum of the proteins to demonstrate that they have purified away any contaminating RNA from the E. coli host.

We recognize the reviewer's concern but we regret that we are not able to provide these spectra, as 1) our Akta FPLC system is equipped with only a 280 nm UV lamp and not 260 nm for RNA absorption. While we cannot rule out any competing co-purified RNA, we would like to draw reviewer's attention that all the proteins used for this assay were purified using the same protocol and, while we cannot rule out the presence of contaminating nucleic acids, the read out in this assay is based on we use of ³²P-labeled RNA substrate. Accordingly, the only signal we observe in our EMSAs is due to the added 5' UTR RNA.

2. Supplemental Data Figure 1 does not include SDS-PAGE gels for the HSUD, ADRP, or N protein samples used in this study.

These have been added to Supp. Data. Fig. 1 (panels b through h were shifted to accommodate) and the legend has been updated accordingly.

3. In Supplemental Data Figure 1b-h, SEC experiments are performed with an analytical column, except for panel 1c, where the retention volume is consistent with a preparative scale column. Perhaps the authors can indicate which column is used for each experiment in the figure or elaborate in the Methods section.

This has been updated in the Materials and Methods (line 466):

"Other purified proteins were similarly purified by size exclusion chromatography on a Superdex 200 HiLoad 10/300 column equilibrated with the same buffer."

4. In Supplemental Figure 1d, the SEC retention volume of the NAB (1089-1203) construct is interpreted as indicating a molecular weight of 14.3 kDa, or a monomeric specie. However, the peak clearly elutes before the 17 kDa marker protein and is closer to the 44 kDa marker. This would indicate the molecular weight to be substantially larger than the monomer. The same issue applies to construct Y4 (1844-1945) in panel 1h.

We thank the reviewer for noticing this. Upon reviewing the Supp. Fig. 1 profiles again, we realized that our labeling of the calibration elution volumes (the arrows in the corresponding figure) was incorrect for the NAB (1089-1203) and Y4 (1844-1945) constructs. We have corrected the position of the arrows in Supp. Fig. 1d and 1h. We

also deleted the following words from the Discussion to be more clear that we believe NAB 1089-1203 to be monomeric: “forms a stable dimer and”

5. In Supplemental Data Table 1, for the Nsp3 NAB (7LGO) crystal, the resolution limits reported under the Data Collection section are not consistent with the limits given in the Refinement section (that is, 2.44 Å data was collected, but the model was refined against 1.74 Å data).

We thank the reviewer for noticing this. Indeed there has been a mishap in the value of resolution and in the number of reflections figures presented for determination of 7LGO structure in this table. To explain, the presented numbers were from another structure we determined for this polypeptide with a resolution of 1.74 Å. However this structural model had some crystallographic issues which prevented us from using it in our analysis. Unfortunately, this led to a mistake while filling the statistics table, which we sincerely apologise for. This has been corrected in the revised submission.

6. In Supplemental Data Table 1, for crystals Nsp3 βSM (7T9W) and Nsp3 Y4 (7RQG), the reported number of unique reflections does not seem consistent. Since both crystals diffract to about the same resolution (~2.2 Å), we would expect to see fewer unique reflections for βSM, since this crystallizes in the much higher symmetry space group P43. Perhaps these numbers represent the total, instead of unique, reflections observed?

We double-checked our data, and no errors were made in the Supp. Data Table 1. The explanation for the large difference in unique reflections is the large difference in size of the asymmetric units (ASU) of these two crystals. Using the Matthews Coefficient calculator at <https://csb.wfu.edu/tools/vmcalc/vm.html>, the unit cell dimensions and the molecular weights, the volumes of ASUs for 7RQG and 7T9W are 107548 and 1709595 Å³, respectively. This is also reflected in the number of protein chains in the asymmetric unit: 7RQG contains 4 polypeptide copies in the ASU and 7T9W contains 16, resulting in many more unique reflections in 7T9W.

7. In the main text, lines 157-158, please report the C α -RMSD between the form 1 and form 2 structures.

We added the following sentence (line 158):

“The RMSDs between the one chain in Ubl1 form 2 and the two chains of Ubl form 1 are 0.23 Å and 0.33 Å over 100 or 101 C α atoms.”

8. In the main text, line 185, the authors may wish to calculate the buried surface area of the interface between the two chains to help gauge if this is a "crystallographic dimer" or is relevant to the dimeric solution structure suggested by the NAB 1050-1216 fragment.

We performed this analysis and updated the main text accordingly (line 188):

“Analysis by PDBePISA on the 1089-1201 crystal structure identified the buried surface area between protomers is 1370 Å³, but does not predict a stable dimer. Further analysis of the NAB is required to advance our understanding of functional relevance of dimerization in the NAB, as ours remains the only experimentally derived structure of this portion of Nsp3 currently available.”

9. In the main text, line 194, can the difference in conformation of these linkers be attributed to differing lattice interactions in the two structures?

We agreed with reviewer’s comment and altered the main text (line 198) to the following:

This difference may be due to these regions serving as flexible linkers to the PIP_{ro} and β SM domains, regions mediating dimerization or differing crystal lattice packing

10. The SEC-RALS experiment shown in Supplemental Data Figure 4 is described in the Materials Methods section but not elaborated on in the Results or Discussion. Does this higher order complex have some biological significance?

We regret not elaborating on the purpose of the SEC-RALS experiment in the Results or Discussion. We performed this experiment to estimate the molecular weight and stoichiometry of the N-Ubl1 complex, which was necessary to calculate the K_d for its interaction with RNA. We updated the Results section to mention this (line 291 in the revised manuscript):

“The EMSA assay against the same substrate using the His₆-Ubl1 and N protein complex shows even higher affinity, reflected in calculated K_D of 7.0 nM \pm 0.8 nM using

an estimated 4:4 complex as suggested by SEC-RALS analysis (**Figure 2d, Supplemental Data Figure 4**).”

We elected not to discuss this further in the Discussion section since we already discussed how oligomerization observed in solution with the isolated proteins may not be relevant in the context of the full-length DMV pore (i.e. line 397 in the revised manuscript).

11. Some of the gel shift assays (for example Figure 3C) reveal activity retained in the pocket of the gel, which is not uncommon for RNA-binding proteins. However, it does complicate the quantification, as these complexes are typically not included in the estimation of the fraction bound. The authors may consider to indicate only a rough estimate of the affinity for these cases.

We considered the bands present at the top of the gel to be aggregated/precipitated protein-RNA complexes. Therefore, we chose to ignore these bands, but we acknowledge the reviewer’s comment that this complicates the estimation of the fraction bound. This is why we chose to show the “approximate” symbol ~ in our figures, which we believe already addresses the suggestion to consider indicating a rough estimate of the affinity.

Errata

1. In main text line 167, the reference to the SEC data should point to Supplemental Data Fig. 1c, not 1b.

Corrected.

2. In main text line 181, "NAR" should be "NAB".

Corrected.

3. In main text line 225, the figure reference should point to Supplemental Data Fig. 2d and 2e.

Corrected to 2d.

Re: Spectrum02871-24R1 (Structural and functional analyses of SARS-CoV-2 Nsp3 and its specific interactions with the 5' UTR of the viral genome)

Dear Dr. Peter J. Stogios:

Your manuscript has been accepted, and I am forwarding it to the ASM production staff for publication. Your paper will first be checked to make sure all elements meet the technical requirements. ASM staff will contact you if anything needs to be revised before copyediting and production can begin. Otherwise, you will be notified when your proofs are ready to be viewed.

Sincerely,
Vaithilingaraja Arumugaswami
Editor
Microbiology Spectrum

Reviewer #1 (Comments for the Author):

The authors have addressed the critiques and substantially clarified the manuscript. I have no additional comments or concerns.